# Therapeutic Potential of Dopamine and Related Drugs as Anti-Inflammatories and Antioxidants in Neuronal and Non-Neuronal Pathologies

**DOI:** 10.3390/pharmaceutics15020693

**Published:** 2023-02-18

**Authors:** Cindy Bandala, Noemi Cárdenas-Rodríguez, Julieta Griselda Mendoza-Torreblanca, Itzel Jatziri Contreras-García, Valentín Martínez-López, Teresita Rocio Cruz-Hernández, Jazmín Carro-Rodríguez, Marco Antonio Vargas-Hernández, Iván Ignacio-Mejía, Alfonso Alfaro-Rodriguez, Eleazar Lara-Padilla

**Affiliations:** 1Neurociencia Básica, Instituto Nacional de Rehabilitación LGII, Secretaría de Salud, Mexico City 14389, Mexico; 2Escuela Superior de Medicina, Instituto Politécnico Nacional, Mexico City 11340, Mexico; 3Laboratorio de Neurociencias, Subdirección de Medicina Experimental, Instituto Nacional de Pediatría, Mexico City 04530, Mexico; 4Laboratorio de Fisiología, Escuela Militar de Graduados de Sanidad, UDEFA, Mexico City 11200, Mexico; 5Unidad de Ingeniería de Tejidos, Terapia Celular y Medicina Regenerativa, Instituto Nacional de Rehabilitación Luis Guillermo Ibarra Ibarra, Mexico City 14389, Mexico; 6Escuela de Biología Experimental, Unidad Iztapalapa, Universidad Autónoma Metropolitana, Mexico City 09340, Mexico; 7Subdirección de Investigación, Escuela Militar de Graduados de Sanidad, Mexico City 11200, Mexico; 8Laboratorio de Medicina Traslacional, Escuela Militar de Graduados de Sanidad, Mexico City 11200, Mexico

**Keywords:** dopamine, anti-inflammatory, antioxidant, antiangiogenic, antinociceptive, nonneurological disease

## Abstract

Dopamine (DA), its derivatives, and dopaminergic drugs are compounds widely used in the management of diseases related to the nervous system. However, DA receptors have been identified in nonneuronal tissues, which has been related to their therapeutic potential in pathologies such as sepsis or septic shock, blood pressure, renal failure, diabetes, and obesity, among others. In addition, DA and dopaminergic drugs have shown anti-inflammatory and antioxidant properties in different kinds of cells. Aim: To compile the mechanism of action of DA and the main dopaminergic drugs and show the findings that support the therapeutic potential of these molecules for the treatment of neurological and non-neurological diseases considering their antioxidant and anti-inflammatory actions. Method: We performed a review article. An exhaustive search for information was carried out in specialized databases such as PubMed, PubChem, ProQuest, EBSCO, Scopus, Science Direct, Web of Science, Bookshelf, DrugBank, Livertox, and Clinical Trials. Results: We showed that DA and dopaminergic drugs have emerged for the management of neuronal and nonneuronal diseases with important therapeutic potential as anti-inflammatories and antioxidants. Conclusions: DA and DA derivatives can be an attractive treatment strategy and a promising approach to slowing the progression of disorders through repositioning.

## 1. Introduction

Dopamine (DA) is a monoamine synthesized mainly in neurons of the midbrain cores, ventral tegmental area, and substantia nigra pars compacta. The synthesis of the neurotransmitter takes place in the dopaminergic nerves [1]. Hydroxylation of the amino acid L-tyrosine is the point of regulation of the synthesis of catecholamines, including DA, in the central nervous system (CNS), and consequently, the tyrosine hydroxylase (TH) enzyme is the limiting enzyme of the synthesis of DA, norepinephrine, and adrenaline. Through their receptors, DA has been shown to have physiological functions in the CNS, such as wakefulness, attention, memory formation and consolidation, novelty-induced memory encoding, and reward/addiction [2,3,4,5]. DA is a neuromodulator that has the ability to diffuse away from the site of its release, activating receptors that are far from the terminal; this ability is called transmission volume [2]. In this sense, DA receptors have been identified in nonneuronal tissues, which has been related to their therapeutic potential in pathologies such as sepsis or septic shock, blood pressure, renal failure, diabetes, and obesity, among others [6,7,8]. In addition, it has been reported that DA and dopaminergic drugs such as bromocriptine, cabergoline, pramipexole, and ropinirole have shown anti-inflammatory and antioxidant functions in different kinds of cells, reducing reactive oxygen species (ROS) accumulation, preserving glutathione (GSH) and other antioxidant enzymes, and decreasing lipid peroxidation [9,10,11,12,13,14]. Additionally, some herbal compounds have shown dopaminergic properties; for example, Hepad S1, a Korean medicinal herbal combination, is an important source of dopamine with neuroprotective properties that improve Parkinson’s symptoms; it could modulate adverse cellular events such as inflammation and oxidation in neuronal cells [15]. Curcumine has shown neuroprotective properties and is an important component of dopamine [16], and Hordenine, a natural compound of germinated barley, is an agonist of the dopamine D2 receptor [17]. These and other herbs have been mainly studied in neuronal diseases, with less research in nonneuronal diseases. Then, the scope of this review is to compile the mechanism of action of DA and the main dopaminergic drugs and show the findings that support the therapeutic potential of these molecules for the treatment of neurological and non-neurological diseases considering their antioxidant and anti-inflammatory properties and their efficacy in clinical assays.

## 2. Methodology

Advanced searches were performed in PubMed, ProQuest, EBSCO, Scopus, Science Direct, Google Scholar, Web of Science, PubChem, NCBI Bookshelf, DrugBank, livertox, and Clinical Trials. We considered the original manuscripts, reviews, minireviews, systematic reviews, meta-analyses, clinical assays, books, and specialized databases. The search was performed by applying the following keywords alone or in combination: “dopamine”, “dopaminergic drug”, “metabolism”, “chemical compounds”, chemical structure”, “D1, D2 receptors”, “precursors”, “experimental agonists and antagonists”, “receptor blockers”, “antioxidant”, “anti-inflammatory”, “neuronal pathologies” “nonneuronal pathologies”, “physiological functions”, “drug repositioning”, “neuromodulator”, “free radicals”, “reactive oxygen species”, “oxidative stresses”, “antioxidant enzymes”, “efficacy”, and “secondary effects”. A total of 200 references were included.

## 3. Dopamine Synthesis, Release, Catabolism, and Postsynaptic Action

In this section, we describe DA and its pharmacological properties at the molecular level. The synthesis of DA (Figure 1) begins with the hydroxylation of L-tyrosine by the TH enzyme to generate L-3,4-dihydroxyphenylalanine (L-DOPA); then, aromatic L-amino acid decarboxylase (AADC or DOPA decarboxylase) allows the production of cytosolic dopamine [18,19,20]. The DA synthesized in the presynaptic terminal is loaded in synaptic vesicles by vesicular monoamine transporter 2 (VMAT-2); subsequently, DA is released to the synaptic cleft. Next, the Na^+^-dependent dopamine transporter (DAT), localized in neurons and glial cells, reuptakes the neurotransmitter [18]. DA is recycled into synaptic vesicles or degraded by specialized enzymes [21], where its catabolism takes place. In presynaptic terminal and glial cells, the monoamine oxidase (MAO) enzyme, localized in mitochondria, breaks down DA through oxidative deamination, producing 3,4-dihydroxyphenylacetaldehyde (DOPAL); in turn, aldehyde dehydrogenase (ALDH) converts DOPAL to carboxylic acid 3,4-dihydroxyphenylacetic acid (DOPAC) by oxidation, or alcohol dehydrogenase (ADH) reduces DOPAL to 3,4-dihydroxyphenylethanol (DOPET) [20,22]. The catechol O-methyl-transferase (COMT) enzyme, localized in the synaptic cleft, catalyzes the methylation of dopamine to 3-methoxytyramine (3-MT), which is a MAO substrate that forms 3-methoxy-4-hydroxyphenylacetaldehyde (HMPAL). Finally, the ALDH enzyme catalyzes HMPAL to generate homovanillic acid (HVA), which is the main end-product of DA degradation [20,22,23]. At the post-synapse, DA binds to D1-like and D2-like receptors, which are G-protein-coupled channels [24]. The D1-like receptor activates the G_αs/olf_ subunit protein that stimulates the adenylyl cyclase (AC) protein; then, it generates the cyclic adenosine monophosphate (cAMP) second messenger, which activates protein kinase A (PKA), resulting in target action and increasing protein phosphorylation. On the other hand, the D2-like receptor, by activating the G_αi/o_ subunit, inhibits the effector protein AC, inhibiting the cAMP second messenger and, thereby, PKA, generating a decrease in protein phosphorylation [18,24,25,26].

## 4. Chemical Compounds and Drugs Related to the Dopaminergic System

There are more than 200 chemical compounds and drugs related to the dopaminergic system [27,28,29], and mentioning each of them is beyond the scope of this work; however, they can be grouped, according to their activity, as precursors [30,31,32,33], agonists and antagonists of receptors [34], DA reuptake inhibitors [35,36] DA releasing agents [36,37], activity enhancers [38,39,40], and enzyme inhibitors [41], among others. Three DA precursors are used in the clinic, L-phenylalanine, L-tyrosine, and L-DOPA; tyrosine is a nonessential amino acid that is synthesized from the essential aromatic amino acid phenylalanine, and both amino acids constitute the two initial steps in the biosynthesis of DA [31]. Levodopa (L-DOPA) is a dopamine precursor and is the most effective and commonly used drug for the treatment of Parkinson’s disease. Levodopa is prescribed in most cases with Carbidopa, which is an inhibitor of L-amino acid decarboxylase, the enzyme that metabolizes levodopa peripherally [42].

DA agonists exert their effects by acting directly on dopamine receptors and mimicking endogenous neurotransmitters. There are two subclasses, ergoline, and nonergoline agonists, with a variable affinity for different DA receptors [43,44]. DA antagonists block the effects of dopamine or its agonists by binding to DA receptors. A variety of DA antagonists are used for the treatment of psychotic disorders; however, their therapeutic effects are mostly due to long-term adjustments rather than acute blockade of DA receptors [29]. Some DA antagonists have been used to treat Tourette’s syndrome or hiccups [45,46], and they have also been used as antiemetics to treat various causes of nausea and vomiting [47]. Table 1 details the mechanisms of action and indications of DA precursors and the most representative dopaminergic agonist and antagonist drugs.

On the other hand, DA reuptake inhibitors may be classified as DAT inhibitors and VMAT inhibitors. The former block the action of DAT, and DA reuptake inhibition occurs when extracellular DA, which does not bind to the postsynaptic neuron, is blocked from re-entering the presynaptic neuron, resulting in increased extracellular concentrations of DA and an increase in dopaminergic neurotransmission [80]. DAT inhibitors are indicated for the treatment of attention deficit hyperactivity disorder, major depressive disorder, and seasonal affective disorder and as an aid to smoking cessation; examples are methylphenidate [27,29]. On the other hand, VMAT inhibitors prevent the reuptake and storage of monoamine neurotransmitters in synaptic vesicles, making them vulnerable to metabolism by cytosolic enzymes. Inhibition of VMAT-2 results in decreased reuptake of monoamines and depletion of their reserves in nerve terminals. They are used to treat chorea due to neurodegenerative diseases or dyskinesias due to neuroleptic medications; examples are tetrabenazine, deutetrabenazine, and valbenazine [27,42,81,82,83].

DA-releasing agents are a type of drug that induces, through various mechanisms, the release of DA from the presynaptic neuron into the synaptic cleft, leading to an increase in extracellular concentrations of the neurotransmitter. Examples are amphetamine, lisdexamfetamine (L-lysine-d-amphetamine; vyvanse), methamphetamine, methylenedioxymethamphetamine (MDMA), and 4-methylaminorex [27,84,85,86,87]. Moreover, (-)1-(benzofuran-2-yl)-2-propylaminopentane, (-)BPAP, (-)-1-phenyl-2-propylaminopentane, and (-)PPAP are enhancers of dopamine activity. BPAP and PPAP act as potent stimulants of neurotransmitter release in dopaminergic neurons, leaving MAO activity largely unchanged. BPAP and PPAP controllably increase the quantity of neurotransmitters that are released when a neuron is stimulated by a neighboring neuron, and they are currently in the research phase [39,88,89].

DA enzyme inhibitors can be classified into DA synthesis inhibitors and DA degradation inhibitors. There are three kinds of dopamine synthesis inhibitors: (1) TH inhibitors (for example, 3-iodo-tyrosine and metyrosine), which are able to inhibit TH activity, the rate-limiting enzyme in DA biosynthesis [90]; (2) phenylalanine hydroxylase inhibitors (for example, 3,4-dihydroxystyrene), which inhibit the enzyme that converts phenylalanine to tyrosine [91]; and (3) DOPA decarboxylase inhibitors, which block the biosynthesis of L-DOPA to DA. Examples of these inhibitors are benserazide and carbidopa, commonly used in combination with levodopa. Since they can hardly cross the blood–brain barrier, they prevent the formation of dopamine in extracerebral tissues, minimizing the occurrence of extracerebral side effects [92,93].

Finally, the main DA degradation inhibitors can be classified into MAO and COMT inhibitors. The most prescribed MAO inhibitors are selegiline, isocarboxazid, phenelzine, and tranylcypromine. They have in common the ability to block oxidative deamination of DA and subsequently provoke its elevation in brain levels, enhancing dopaminergic activity [29,94]. Selegiline is close structurally to (-) methamphetamine and is a selective and irreversible inhibitor of monoamine oxidase type B (MAO-B). Selegiline is the first catecholaminergic activity-enhancing substance in clinical use that does not continually release catecholamines and is, therefore, free of amphetamine dependence [38,40]. Likewise, the most common COMT inhibitors are entacapone, opicapone, and tolcapone. They inhibit the COMT enzyme and are frequently used in the treatment of Parkinson’s disease as an adjunct to levodopa/carbidopa medication [95,96,97]. Many Parkinson’s disease patients treated with levodopa plus carbidopa experience motor complications over time; when COMT inhibitors are administered, plasma levodopa levels are increased and maintained, resulting in more consistent dopaminergic stimulation, leading to further reduction of the manifestations of parkinsonian syndrome [98].

In summary, dopaminergic compounds and drugs act through a variety of mechanisms of action within the process of synthesis, release, catabolism, and postsynaptic action of dopamine, as shown in Figure 2. It should be noted that these main mechanisms are often accompanied by secondary mechanisms (such as antioxidant or anti-inflammatory mechanisms, see below, which are not yet fully understood) that give a wide variety of effects and indications as potential adjuvants in most chronic and degenerative diseases.

## 5. Antioxidant and Anti-Inflammatory Properties of Dopamine and Related Drugs

In 1997, it was reported for the first time that DA has a direct antioxidant effect due to the number of hydroxy groups on the phenolic ring of the molecule. In this sense, Yen and Hsieh [99] showed that DA has a protective effect against the oxidation of linoleic acid, has reducing power, and shows scavenger capacity against 1,1-diphenyl-2-picryl-hydrazyl (DPPH) radicals, superoxide radicals (O_2_^•−^) and hydroxyl radicals (HO•) (94.94, 53 and 65.7%, respectively), showing the strongest capacity. The authors conclude that the 1,2 position hydroxy group on the phenolic ring and the side chain is an electron-donating amine group [99]. Later, it was shown that DA and D4 receptors induced nuclear factor-erythroid 2 related factor 2 (Nrf2) activity during ischemia in vivo in astrocyte and meningeal cell cultures, showing its capacity to modulate the antioxidant effect; Nrf2 is a transcription factor that controls inducible expression of multiple antioxidant/detoxification genes [100,101] and induces the expression of heme oxygenase-1 (HO-1) by human endothelial cells in vitro [102]. The anti-inflammatory effect of DA has also been demonstrated in alcoholic hemorrhagic pancreatitis in cats [103]. In fact, DA has been proposed as an immune transmitter, given that dopaminergic signaling is involved in neurological diseases and is associated with the inflammatory response [104]. DA inhibits cytokine production via D1 receptors, decreases oxidative stress [105], and can cause nuclear factor kappa B (NF-kB), a transcription factor that mediates the control of ROS and inhibition in acute kidney injury [106]. Catecholamines, including DA, can inhibit tumor necrosis factor-alpha (TNF-α) and may enhance interleukin-6 (IL-6) and interleukin-10 (IL-10) production through D2, D3, or D1/D5 receptors [107,108,109]. In fact, DA has been proposed to be a putative anti-inflammatory cytokine by itself attenuating the chemoattractant effect of interleukin-8 (IL-8), integrins CD11b and CD18, and the adhesion molecules E-selectin and intercellular adhesion molecule 1 (ICAM-1) [110]. DA and its D1 receptor also inhibit the activation of the protein complex named NLR family pyrin domain containing 3 (NLRP3) inflammasome in bone marrow-derived macrophages [111,112]. On the other hand, it has been shown that catechol moieties protect cells against oxidative damage and downregulate the pro-inflammatory cytokine interleukine-1beta (IL-1β) in human bone marrow mesenchymal stem cells [113]. Catecholamines identified in two medicinal plants (*Santolina chamaecyparissus* and *Launaea mucronate*) have also shown antioxidant and anti-inflammatory effects in carrageenan-induced paw edema in a rat model [114]. DA also inhibited the peroxidation of brain phospholipids and reaction with radicals such as trichloromethyl peroxyl radicals (CCl_3_O_2_•), O_2_^•−^, peroxynitrite (ONOO^−^) and hydrochlorous acid (HOCl) generated in vitro [115,116,117]. Moreover, the antioxidant effect of DA derivatives of several plant species, such as soybean, avocado, apple, cucumber, and banana, has also been reported, showing an increase in antioxidant enzyme activities (superoxide dismutase, SOD; catalase, CAT; glutathione reductase, GR) and reactive oxygen species (hydrogen peroxide, H_2_O_2_, nitric oxide, NO•, and O_2_^•−^) scavenging capacity [118,119,120,121,122,123]. In another work, it was shown that other derivatives of DA-related compounds or DA agonists showed antioxidant activity. It has been shown that phenolic sulfonamides showed scavenger capacity in vitro against DPPH, 2,2′-azino-bis(3-ethylbenzthiazoline-6-sulfonic acid (ABTS), and O_2_^•−^ [124]. N-Nicotinoyl dopamine also showed antioxidant properties with DPPH scavenging activity and protected against ROS accumulation induced by UVB irradiation in HaCat cells [125]. Bromocriptine, a DA agonist, scavenged O_2_^•−^, 5,5-dimethyl-1-pyrroline-N-oxide hydroxide, and DPPH radicals generated through in vitro systems [9]. This compound also activates NAD(P)H quinone oxidoreductase 1 (NQO1) via Nrf2-phosphatidylinositol-3-kinase/protein kinase B (PI3K/AKT) signaling in H_2_O_2_-treated PC12 cells, protecting against oxidative damage [126]. In in vivo experimental work, it was shown that a non-ergot DA agonist named ropinirole showed a neuroprotective effect, increased GSH, CAT, and SOD antioxidant activities in the striatum, protected striatal dopaminergic neurons against 6-hydroxydopamine (6-OHDA) in mice [14] and was an activator of the GHS system in the mouse striatum [127]. Pramipexole, a DA agonist, protects the DAergic cell line MES 23.5 against 6-OHDA and H_2_O_2,_ increasing cellular levels of GSH, glutathione peroxidase (GPx), and CAT activities [11,12] and scavenging HO• induced by 6-OHDA in rats [13]. In an in vivo model using [3H] pramipexole, it has been shown that the drug enters and accumulates in cells and mitochondria. Pramipexole also prolongs survival time in SOD-1-G93A mice, a model of familial amyotrophic lateral sclerosis [128]. Cabergoline, an ergot derivative DA agonist, has the ability to activate GSH, CAT, and SOD against the neurotoxicity of 6-OHDA in mice, reducing lipid peroxidation [10] and showing antioxidant activity against oxidation in phosphatidylcholine liposomes [129]. D-390, a novel D2/D3 receptor agonist, also showed potent iron chelation [130], and a new tris (DA) derivative also showed Fe(III), Mg(II), Zn(II), and Fe(II) chelation and antioxidant activity in neuron-like rat pheochromocytoma cells [131]. Other DA derivatives, such as N-arachidonoyl-DA and apomorphine, and DA-related compounds, such as pukateine [(R)-11-hydroxy-1,2-methylenedioxyaporphine], have also shown antioxidant properties [60,132,133,134]. It has been shown that caffeic acid anilides and caffeic acid dopamine amide showed DPPH scavenging capacity and microsomal lipid peroxidation-inhibiting activity [135]. Recently, the water-soluble caffeic acid-DA hydrochloride complex has been proposed as a bactericidal, antibiofilm, and antitumoral agent in the physiological pH range (5.5–7.5) due to its antioxidant properties [136].

Recent clinical research findings indicate that melatonin may modulate dopaminergic pathways involved in movement disorders in humans. It has been proposed that the interaction of melatonin with the dopaminergic system may play a significant role in the nonphotic and photic entrainment of the biological clock as well as in the fine-tuning of motor coordination in the striatum principally because these interactions, by its antioxidant nature can be beneficial in humans [137,138,139]. Additionally, its anti-inflammatory properties have been proposed for the treatment of inflammatory bowel disease, rheumatoid arthritis, systemic lupus erythematosus, and multiple sclerosis [140,141]. In relation to pathologies not related to the central nervous system, the use of DA-melanin nanoparticles has been proposed as a novel scavenger of ROS and reactive nitrogen species (RNS). These nanoparticles showed low cytotoxicity and a strong ability to scavenge ROS and RNS: O_2_^•−^, HO• radicals, and ONOO^−^ were proposed as potent anti-inflammatory and chondroprotective agents due to their average diameter of 112.5 nm. Nanoparticles can be intra-articularly injected into an affected joint and retained at the injection site, as was shown in an osteoarthritis rodent model and in chondrocyte cultures. These nanoparticles also diminished IL-1β and reduced proteoglycan loss, probably stimulating autophagy for chondrocyte protection. IL-1β caused an increase in the gene expression of autophagy markers: protein 1A/1B-light chain 3 (LC3-11), autophagy-related 7 (ATG7), and beclin-1 [142]. The use of N-acyl dopamine derivates has also been proposed as a potential alternative for implementation in transplantation medicine due to its immunomodulatory, cytoprotective, and anti-inflammatory properties [143]. The antioxidant and anti-inflammatory properties of DA and some related drugs are summarized in Figure 3.

Finally, it is important to mention that in cancer, DA agonists inhibit T-cell proliferation and cytotoxicity, probably through activation of the D1 receptor, which promotes an increase in intracellular cAMP, contributing to immune regulation [144]. Additionally, these agonists have an important role due to their beneficial antiangiogenic effects. Hoeppner et al., 2015 [145] showed that D2 receptor agonists inhibit NADPH oxidase activity, reducing the production of ROS involved in angiogenesis [145]. Leng et al., 2017 [146] found in GH3 cells that D5 receptor agonists could inhibit the activity and expression of SOD-1 and increase ROS, promoting autophagy and cell death by inhibiting the AKT-mammalian target of rapamycin (mTOR) pathway [146].

## 6. Clinical Trials in Nonneuronal Pathologies

DA, agonists, or derivatives are being tested as possible drugs or adjuvants in other non-CNS pathologies, possibly due to their antioxidant or anti-inflammatory/immunomodulatory properties. In this sense, DA, serotonin, prostaglandin E2, substance P, and lipoperoxidation levels are decreased, whereas SOD levels are increased after pain treatment with warm acupuncture and meloxicam in patients with knee osteoarthritis, showing the involvement of these biochemical markers as anti-inflammatory mediators [147]. DA treatment (15 μg/kg/min) is also effective in increasing blood pressure in neonates with hypothermia treatment for hypoxic-ischemic encephalopathy [148], and the use of the DA synthetic analog dopexamine in doses of 0.5 and 2.0 μg/kg/min significantly protected the upper gastrointestinal mucosa in the of patients with abdominal surgery, reducing the incidence of acute inflammation and decreasing myeloperoxidase activity and inducible nitric oxide synthase in biopsies [149]. The effects of DA (2.5 to 10 μg/kg/min) have also been observed in patients with sepsis, where its administration was associated with a fall in lactate and no effect on arterial pH [150]. DA (10 to 25 µg/kg/min) is effective in the treatment of patients with hyperdynamic septic shock, where it successfully improved the systemic vascular resistance index, cardiac index, oxygen delivery and uptake [151]. It has been shown that DA (infused at 2 and 4 µg/kg/min) increases renal oxygenation with no increase in tubular sodium reabsorption or renal oxygen consumption in glomerular filtration rate in postcardiac surgery patients [152]. Bromocriptine has also been proposed as an adjuvant in immunosuppression after renal transplantation, but its effectiveness has not yet been widely shown [153,154]. Additionally, bromocriptine (2.5 mg twice daily) prevented ulcer relapse for six months in patients with duodenal ulcers [155]. The use of pramipexole (from 0.25 to 0.75 mg) has shown efficacy in the treatment of restless legs syndrome in patients [156,157]. The use of cabergoline (0.5 mg for eight days) and bromocriptine (2.5 mg for 16 days) are efficient in the prevention of moderate and early-onset ovarian hyperstimulation syndrome in patients [158]. The role of DA in crucial social role decision-making was shown using pramipexole in women, allowing them to become less generous in general, modulate smoking behavior or produce subjective effects of cocaine, improve sleep behavior disorder and tinnitus, and help against pain, fatigue, function, and global status in patients with fibromyalgia [159,160,161,162,163,164,165]. Finally, Table 2 summarizes diverse clinical trials in progress.

## 7. Discussion and Conclusions

This is an important work in which the applications of DA and its derivatives are reviewed, offering physicians and healthcare personnel information that may be valuable to make therapeutic decisions considering the advances in the field of knowledge of the use of drugs (of natural or synthetic origin) and/or their action targets. In the present work, we showed that DA and dopaminergic drugs have emerged for the management of diseases, mainly at the neuronal level; however, they have been proposed for the treatment of pathologies that are not directly related to the nervous system, possibly due to their anti-inflammatory and antioxidant properties. *Cabergoline*, *fenoldopam*, *bromocriptine*, *domperidone*, *pramipexole*, *rotigotine*, and *quinagolide*, among others, are being tested for sepsis or septic shock, renal failure, gastric diseases, cancer, brain trauma injury, blood pressure, and fibromyalgia. DA receptor agonists or antagonists can function through classical G protein signaling regulating AKT/NF-κB, rat sarcoma virus (Ras)/PI3K/AKT, cAMP-response element binding protein (CREB)/NF-κB or signal transducers and activators of transcription (STAT) pathways inhibiting or activating nuclear transcription or downstream related factors such as NRLP3 inflammasome expression, mTOR, Nrf2 or a tool-like receptor (TLR). Additionally, they can function through other nonreceptor-dependent pathways as L-type Ca^2+^ channels. However, DA and related drugs should be further studied to more precisely understand the molecular and biochemical mechanisms underlying the large number of therapeutic effects considered in this review. Moreover, because DA receptors have multiple physiological roles in neurological and systemic diseases, more preclinical studies are necessary to elucidate the specific functions of DA receptor subtypes.

On the other hand, considering that many systemic and neurodegenerative diseases are characterized by the presence of inflammation, related in turn to oxidative stress, DA and DA derivatives can be an attractive option as a strategy of treatment and a promising approach to slowing the progression of disorders through the repositioning of DA. In this sense, our review is important since we mention the possible mechanisms by which DA and its derivatives act as anti-inflammatory and antioxidant compounds in in-vitro studies, animal models, and clinical trials where their therapeutic application is being tested.

Furthermore, it is necessary to study natural products containing DA. In this review, some products, such as fruits, vegetables, and plants with dopaminergic content, have shown antioxidant or anti-inflammatory properties. In the literature, active metabolites such as stepholidine (in Chinese herb), pukatein (natural aporphine derivative), salsolinol (in bananas), hordenine (a constituent of barley and beer), goitrin (in brassicaceous weeds), bromophenols curcumin or cannabinoids that showed dopaminergic properties due to the interaction with DA receptors modulating its signaling are also being considered as possible therapeutic agents. In relation to products of natural origin, first, experimental studies are necessary to understand the dynamic behavior of DA receptors and their interaction modes with active metabolites to understand the relevant structural and functional characteristics of these receptors for interaction with metabolites that function as agonists, antagonists or blockers. Second, more experimental and clinical studies are needed to establish which products of natural origin can be used for the treatment of non-neurological diseases related to DA metabolism.

Due to the above, one of the limitations of this work is the lack of knowledge in a deeper and more precise way of the signal transduction mechanisms of DA, related drugs, and natural compounds, considering the physiopathology of the different diseases where they have been applied. In addition, understanding these mechanisms could generate new applications for DA and its derivatives in other diseases and even be considered adjuvants for combined therapies for different types of neuronal and nonneuronal pathologies.

## Figures and Tables

**Figure 1 pharmaceutics-15-00693-f001:**
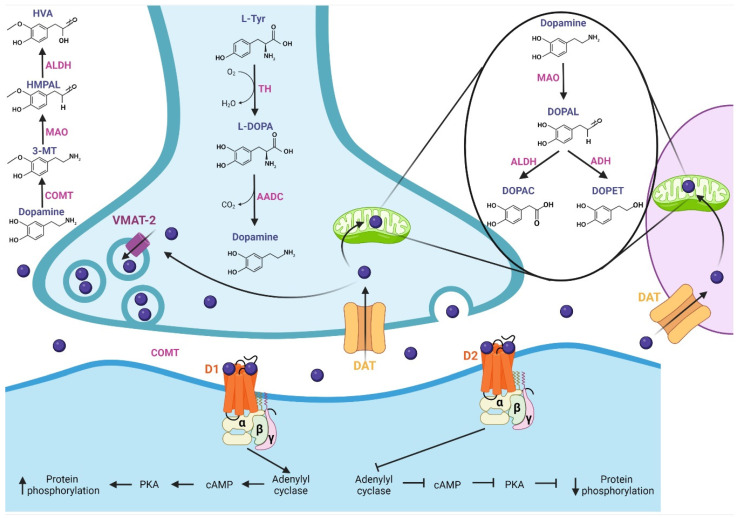
Synthesis, release, catabolism, and postsynaptic action of dopamine. Synthesis: The TH enzyme converts L-tyrosine to L-DOPA; then, the AADC enzyme allows the production of dopamine, which is loaded into synaptic vesicles by VMAT-2. Release and recycling: once released in the synaptic cleft, the DAT transporter reuptakes dopamine, which is recycled into synaptic vesicles. Catabolism: Dopamine is degraded by specialized enzymes; the MAO enzyme breaks down dopamine to DOPAC and DOPET. In the synaptic cleft, the COMT enzyme catalyzes dopamine to HVA, which is the main end-product of dopamine degradation. At the post-synapse, dopamine binds with D1-like and D2-like receptors. The D1-like receptor activates the G_αs/olf_ subunit, which stimulates adenylyl cyclase protein, increasing protein phosphorylation. D2-like receptor, by activating the G_αi/o_ subunit, inhibits the protein adenylyl cyclase, generating a decrease in protein phosphorylation. TH: tyrosine hydroxylase, L-DOPA: L-3,4-dihydroxyphenylalanine, AADC: L-amino acid decarboxylase, VMAT-2: vesicular monoamine transporter 2, DAT: dopamine transporter, MAO: monoamine oxidase, DOPAL: 3,4-dihydroxyphenylacetaldehyde, ALDH: aldehyde dehydrogenase, DOPAC: 3,4-dihydroxyphenylacetic acid, ADH: alcohol dehydrogenase, DOPET: 3,4-dihydroxyphenylethanol, COMT: catechol O-methyl-transferase, HMPAL: 3-methoxy-4-hydroxyphenylacetaldehyde, HVA: homovanillic acid.

**Figure 2 pharmaceutics-15-00693-f002:**
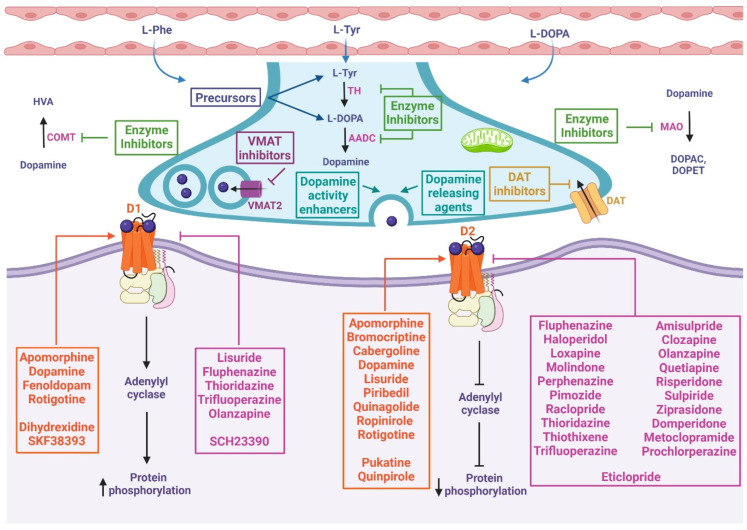
Mechanism of action of chemical compounds and drugs related to the dopaminergic system. These drugs can inhibit or activate diverse proteins involved in dopamine metabolism, including precursors, enzyme inhibitors, dopamine-releasing agents, dopamine reuptake inhibitors, dopamine activity enhancers, and agonists or antagonists of D1-like and D2-like receptors. *At presynapses*: The precursors enable the biosynthesis of dopamine. VMAT inhibitors prevent the storage of monoamines in synaptic vesicles, resulting in the depletion of these neurotransmitters. DAT inhibitors keep dopamine in the synaptic cleft longer by inhibiting its reuptake. Dopamine-releasing agents and dopamine activity enhancers increase the release of the activity of dopamine into the synaptic cleft. Dopamine synthesis inhibitors prevent the formation of dopamine as an endpoint. Dopamine degradation inhibitors enhance dopaminergic activity by blocking dopamine catabolism. *At postsynapses*: The dopamine agonist (orange box) mimics endogenous dopamine function, thus activating or inhibiting adenylyl cyclase depending on whether it binds to D1-like or D2-like receptors, respectively. Dopamine antagonists (pink box) bind to but do not activate dopamine receptors, thereby blocking the actions of dopamine. L-Phe: L-phenylalanine, L-Tyr: L-tyrosine, TH: tyrosine hydroxylase, L-DOPA: L-3,4-dihydroxyphenylalanine, AADC: L-amino acid decarboxylase, VMAT2: vesicular monoamine transporter 2, DAT: dopamine transporter, MAO: monoamine oxidase, DOPAC: 3,4-dihydroxyphenylacetic acid, DOPET: 3,4-dihydroxyphenylethanol, COMT: catechol O-methyl-transferase, HVA: homovanillic acid, D1: dopamine 1 receptor, D2: dopamine 2 receptor.

**Figure 3 pharmaceutics-15-00693-f003:**
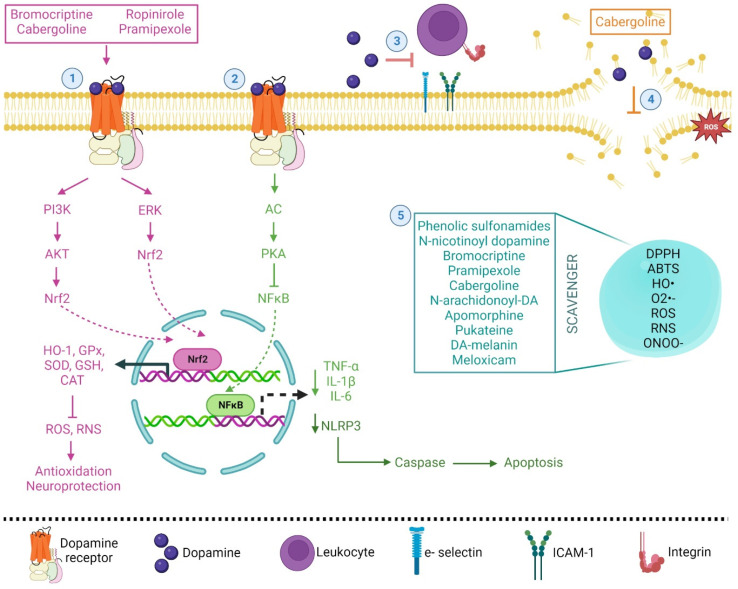
Simplified and integrated mechanisms of dopamine and its agonist, as antioxidant and anti-inflammatory molecules, in various physiological processes. (1) The binding of dopamine and some agonist drugs with its receptor activates the PI3K/AKT or ERK signaling pathway, resulting in Nrf2 translocation to the nucleus, inducing the expression of HO-1 and antioxidant genes (pink pathway). (2) The activation of AC/PKA inhibits NF-κB, generating a decrease in proinflammatory cytokine expression and the protein complex NLRP3 inflammasome, thus diminishing the apoptotic process (green pathway). (3) Dopamine attenuates the chemoattractant effect of integrins and adhesion molecules. (4) Cabergoline and dopamine inhibited the peroxidation of brain phospholipids and reacted with free radicals. (5) Some drugs showed scavenger capacity and protection against ROS accumulation (cyan box).

**Table 1 pharmaceutics-15-00693-t001:** Mechanism of action and indications of dopamine precursors and dopaminergic agonist and antagonist drugs.

Drug	Mechanism of Action	Indications
Precursors		
Levodopa (L-DOPA)	Levodopa mimics the role of endogenous dopamine; crosses the blood–brain barrier through various pathways, and is decarboxylated to form dopamine stimulating the dopamine receptors	Parkinson’s disease [29,30]
L-phenylalanine	Precursor of tyrosine, dopamine, norepinephrine (noradrenaline) and epinephrine (adrenaline)	Antidepressant effects Vitiligo [31,32]
L-tyrosine	Precursor of dopamine, norepinephrine and epinephrine	Antidepressant [31,33]
Receptor agonists		
Apomorphine	A nonergoline dopamine agonist with binding affinity to dopamine D2, D3, and D5 receptors	Parkinson’s disease [29,42,48]
Bromocriptine	Ergoline derivative with strong agonist activity on the D2 dopamine receptors	Parkinsonian Syndrome Amenorrhea Galactorrhea Acromegaly Premenstrual syndrome Female infertility [29,42]
Cabergoline	Ergoline derivative; dopamine agonist (with a high affinity for D2 receptors) and prolactin inhibitor.	Hyperprolactinemic disorders. Parkinsonian Syndrome [29,49,50]
DA	Agonist to the D1, D2, D3, D4, D5 dopamine receptors. Interacts on the synaptic terminals, causing neuronal excitation or inhibition at the target neuron	Hemodynamic imbalances Blood pressure Hypotension Poor perfusion of vital organs Low cardiac output [29,51]
Fenoldopam	Benzazepine derivative; selective dopamine D1 receptor agonist. Decreases peripheral vascular resistance in renal capillary beds	Hypertension [29,52]
Lisuride	Ergoline derivative, agonist to dopamine D2 receptors. It can be an antagonist to dopamine D1 receptors. Additionally, activates some serotonin receptors	Parkinson’s disease [27,29]
Piribedil	Nonergoline, piperazine derivative, dopamine agonist that acts on D2 and D3 receptors	Parkinson’s disease [53]
Pramipexole	Nonergoline, a dopamine agonist showing specificity and strong activity at dopamine D2 receptors	Parkinson’s disease Restless legs syndrome [54,55]
Quinagolide	Nonergoline derivative; it selectively binds to D2 receptors on the surface of lactotroph cells, resulting in reduced adenylyl cyclase activity and inhibition of prolactin secretion from the anterior pituitary	Hyperprolactinemia [27,29]
Ropinirole	Nonergoline derivative, selectively binds to dopamine D2 receptors, with highest affinity at D3 receptors	Parkinson’s disease Restless legs syndrome [56,57]
Rotigotine	Nonergoline derivative; a nonselective agonist of dopamine receptors with higher affinity at D3 receptors	Parkinson’s disease Restless legs syndrome [29,42]
Experimental agonists		
Dihydrexidine (LS-186,899)	Selective full agonist at the dopamine D1 receptors. It has some affinity for the D2 receptor	Scientific research [58,59]
Pukateine	Aporphine derivative; agonist at the D2 dopamine receptor and antagonist at the α1 adrenergic receptor	Scientific research [60,61]
Quinpirole	Selective D2 and D3 receptor agonist	Scientific research [62,63]
SKF 38393	A selective D1-like receptor agonist	Scientific research [64]
Antagonists and receptor blockers		
Typical antipsychotics		
Chlorpromazine	Phenothiazine derivative. It binds strongly to the D2 receptor, blocking its action; this blockade, in the nigrostriatal pathway, is responsible for its extrapyramidal side effects	Schizophrenia, Bipolar disorder Acute psychosis, Nausea and vomiting Relief of apprehension before surgery Persistent singultus (chronic hiccups) [29,46,65]
Fluphenazine	Phenothiazine derivative. Blocks postsynaptic mesolimbic dopaminergic D2 receptors in the brain	Management of psychosis in schizophrenia [29,66]
Haloperidol	It is a first-generation antipsychotic and one of the most frequently used worldwide. It is not selective for the D2 receptor, but, has a strong antagonism to this dopamine receptor in mesolimbic and mesocortical pathways in the brain	Schizophrenia Tourette syndrome Behavioral disorders in children Hyperactivity [29,45]
Loxapine	Dibenzoazepine tricyclic derivative. Antagonist with high affinity for the D2 receptor, also a serotonin 5-HT2 blocker	Schizophrenia Other psychotic disorders [27,67]
Molindone	Indole derivative. Antagonizes dopamine D2 receptor sites in the reticular limbic systems in the brain, decreasing dopamine activity	Schizophrenia [27,29]
Perphenazine	Phenothiazine derivative. It binds to the dopamine D1 and dopamine D2 receptors inhibiting their activity. Its antiemetic effect is mainly due to blockade of D2 receptors in the chemoreceptor trigger zone and the vomiting center	Schizophrenia Other psychotic disorders Nausea and vomiting [27,29]
Pimozide	Diphenylbutylpiperidine derivative. It binds and inhibits the dopamine D2 receptor	Tourette syndrome Schizophrenia [27,42]
Thioridazine	Phenothiazine derivative. It blocks postsynaptic mesolimbic dopaminergic D1 and D2 receptors	Schizophrenia Other psychotic disorders Generalized anxiety disorder Depressive disorders [27,68]
Thiothixene	Thioxanthene derivative. Is a highly potent antagonist of the D1, D2, D3 and D4 dopamine receptors.	Schizophrenia [27,29]
Atypical antipsychotics		
Amisulpride	Benzamide derivative. It is a selective dopamine D2 and D3 receptor antagonist	Schizophrenia Nausea and vomiting [27,29]
Clozapine	Dibenzodiazepine derivative. It binds to the D4 dopamine receptors with a higher affinity than D2 receptor. Additionally, it has antagonistic effects at 5-HT2A receptors and is a partial agonist at 5-HT1A receptors	Resistant schizophrenia [27,69]
Olanzapine	Thienobenzodiazepine derivative. It exerts its action primarily on dopamine D1, D2, D3 and D4 and serotonin 5-HT2A, 5-HT2C, 5-HT3 and 5-HT6 receptors, blocking their action	Schizophrenia, Bipolar disorder [27,70]
Quetiapine	Thiazepine derivative. Antagonizes to D2 dopamine receptors and to 5-HT2A receptors (it has strong affinity for the latter)	Schizophrenia Bipolar disorder Major depressive disorder [27,71]
Risperidone	Benzisoxazole derivative. It blocks D2, but more 5-HT2A receptors in the brain.	Schizophrenia, Bipolar mania Autism-associated irritability [29,72]
Sulpiride	Benzamide derivative. Selective antagonist at dopamine D2, D3 receptors	Schizophrenia [27,29]
Ziprasidone	Benzothiazolylpiperazine derivative. Binds to 5-HT2A and dopamine D2 receptors with high affinity	Schizophrenia Bipolar mania Acute agitation in schizophrenic patients [29,73]
Antiemetics		
Domperidone	Benzimidazole derivative. It has strong affinity for the D2 and D3 dopamine receptors, blocking their activity	Peristaltic stimulant, Dyspepsia, Indigestion, Epigastric pain Nausea and vomiting [27,29,74]
Metoclopramide	Benzamide derivative. Inhibit dopamine D2 and serotonin 5-HT3 receptors	Nausea and vomiting Gastroesophageal reflux disease [29,75]
Prochlorperazine	Phenothiazine derivative. It mainly blocks the D2 dopamine receptors in the mesolimbic system	Schizophrenia, Schizoaffective Other conditions with psychosis symptoms Nausea and vomiting [29,76]
Experimental antagonists		
Eticlopride	Antagonizes D2 dopamine receptor	Scientific research [77]
Raclopride	Potent and selective antagonist on D2/D3 dopamine receptors	Trials studying Parkinson’s disease [29,78]
SCH23390	Highly potent and selective of D1 dopamine receptor	Scientific research of drug addiction [79]

**Table 2 pharmaceutics-15-00693-t002:** Clinical trials where the effects of DA and DA agonists or derivatives are being studied in non-CNS diseases as possible drugs or adjuvants.

Intervention	Condition or Disease	Study Design and Location	Outcome Measures
Sepsis or septic shock			
DA (5–20 μg/kg/min to predetermined max of 20)	Septic shock	Interventional, randomized, parallel, assignment N = 252 Location: United States	Efficacy, safety, arrhythmia (28 days) [166]
DA (Start at 5 μg/kg/min, increase every 16–30 min by 5 μg/kg/min to a maximum dose of 15 μg/kg/min or adequate response)	Late-onset neonatal sepsis Extreme prematurity neonatal hypotension	Observational, prospective N = 550 Location: Canada	Mortality, severe neurological injury (assessed up to a maximum of 36 weeks after date of birth) Treatment failure rate (Time frame: 90 min) Bronchopulmonary dysplasia, retinopathy, and prematurity (Time: frame: 36 weeks postmenstrual age) Length of hospital stay [167]
DA (8 μg/kg/min, increasing the dose after 15 min to 12 μg/kg/min to a maximum of 15 μg/kg/min)	Shock Hypovolemic Septic Shock	Interventional, randomized, parallel, assignment N = 135 Location: Bangladesh	Case fatality rate (Time frame: 28 days) Treatment failure rates, need of mechanical ventilation, heart failure, length of ICU stay and heart function (Time frame: 7 days) [168]
Renal failure			
Fenoldopam (D1 agonist)	Acute renal failure	Interventional, randomized, single group assignment Location: United States	Incidence of death or dialysis at 21 days Peak serum creatinine and duration of ICU stay [169]
DA (dose of 4 μg/kg/min to the renal graft donor after induction of anesthesia till ligation of the renal artery)	Renal failure Transplant renal failure	Interventional, randomized, parallel assignment N = 60 Location: Egypt	Post-operative creatinine clearance (Time frame: 7 days) [170]
Ropinirole (0.50 mg capsule, once daily for 4 weeks)	End stage renal disease	Interventional, randomized, crossover Assignment N = 52 Location: Canada	Quality of life scale patient global impressions (Time frame: 18 weeks) [171]
Fenoldopam. 60 μg/mL; 0.1 mL/h to provide 0.1 μg/kg/min). If, after 6 h there is not a clinically concerning decrease in blood pressure, as determined by attending physician, the rate of infusion will be increased to 0.2 mL/kg/h (0.2 μg/kg/min for infants receiving fenoldopam). This rate will be continued throughout the remainder of the study	Acute kidney injury Patent ductus arteriosus	Interventional, randomized, parallel assignment N = 1 Location: United States	Changes in urine output (mL/kg/h) and serum levels of fenoldopam during infusion of the drug and following discontinuation of the drug will be measured by liquid chromatography and mass spectrometry (Time frame: 60 h) Change in levels of serum albumin, β-2 macroglobulin, epidermal growth factor, osteopontin, uromodulin cystatin C and in serum creatinine (mg/dL) (Time frame: 48 h) [172]
Fenoldopam (continuous intravenous infusion of 0.3 μg/kg/min fenoldopam for 24 h)	Acute renal failure	Interventional, randomized, parallel assignment N = 80 Location: Poland	Cystatin C and Neutrophil Gelatinase-Associated Lipocalin (NGAL) in serum (Time frame: after 24 and 48 h) [173]
Fenoldopam (continuous infusion at 1 μg/kg/min during cardiopulmonary bypass)	Acute renal failure	Interventional, randomized, parallel assignment N = 80 Location: Italia	Reduction of urinary and/or serum levels of biomarker NGAL in treated group versus controls Reduction of urinary and/or serum levels of cystatin C, increase of diuresis and improvement of perfusion markers in treated group versus controls (Time frame: end of surgery and 12 h postoperatively) [174]
Fenoldopam. 0.1 μg/kg/min (from 0.025 to 0.3 μg/kg/min) for up to 4 days	Acute renal failure	Interventional, randomized, parallel assignment N = 667 Location: Italia	Number of patients requiring Renal Replacement Therapy (Time frame: participants will be followed for the duration of intensive care unit stay, an expected average of one week). Number of dead patients (Time frame: Participants will be followed for 1 year) [175]
Gastric diseases			
Domperidone (DA agonist)(10–30 mg oral dose, four times daily)	Gastroesophageal reflux Gastroparesis Chronic constipation	Expanded Access Location: United States	Unspecified data [176]
Domperidone (10 mg administered 2–4 times a day as needed)	Gastroparesis Esophagitis Dyspepsia Chronic idiopathic Constipation Nausea Vomiting	Interventional, randomized, single group assignment N = 42 Location: United States	Relief for patients with gastrointestinal disorders who have failed standard therapy (Time frame: if the subjects continue to take domperidone) [177]
Disease in ovarian and Cushing’s disease			
Cabergoline (0.5 mg/day for 8 days)	Polycystic ovarian syndrome	Interventional, nonrandomized, single group assignment N = 40 Location: Turkey	Concentrations of follicular fluid antimullerian hormone (Time frame: 1 year) [178]
Quinagolide (DA Agonist) (200 μg/day)	Ovarian hyperstimulation syndrome	Interventional, randomized, parallel assignment N = 30 Location: Spain	Tolerability of quinagolide 200 μg/day in a dose-titration regimen in oocyte donors undergoing controlled ovarian hyperstimulation and at risk of developing ovarian hyperstimulation syndrome (Time frame: 21 days) [179]
Cabergoline (1 mg/week in divided doses, increased by 1 mg/week every month, to the maximum of 5 mg/week. If response is seen than the dose at which response is seen is continued until the end of the study)	Cushing’s disease	Interventional, nonrandomized, single group assignment Location: India	Response in term of mid night cortisol < 5.0 μg/dL and/or Standard two-day dexamethasone suppression test < 1.8 μg/dL [180]
Diabetes			
Bromocriptine	Type 1 diabetes Cardiovascular disease	Interventional, randomized, crossover assignment N = 108 Location: United States	Mean glucose, insulin dosing, brachial artery distensibility, hyperemia, peripheral arterial tonometry (Time frame: 4 weeks) [181]
Bromocriptine (1.6–3.2 mg/day)	Diabetes autonomic neuropathy	Interventional, randomized, parallel assignment N = 84 Location: United States	Changes in expiration/inspiration ratio, in bromocriptine ratio, in electrochemical skin conductance and in heart rate variability (Time frame: 24 weeks) [182]
Cabergoline (0.5 mg per week)	Diabetes type 2	Interventional, randomized, single group assignment N = 10 Location: Iran	Fasting blood sugar and glycosylated hemoglobin (Time frame: 30 days) [183]
Bromocriptine (2.4–3.2 mg/day)	Diabetes type 2	Interventional, randomized, single group assignment N = 23 Location: United States	Glucose metabolism during mixed meal tolerance test (Time frame: 5 weeks) [184]
Bromocriptine	Insulin sensitivity	Interventional, randomized, crossover assignment N = 15 Location: Netherlands	Timing of administration of bromocriptine. Difference in insulin sensitivity between lean and obese males before and after the use of bromocriptine Difference in energy expenditure in lean and obese before and after the use of bromocriptine. (Time frame: 6 weeks) [185]
Obesity or overweight			
Bromocriptine (1.6 mg)	Obesity and overweight Eating behavior	Interventional, randomized, crossover assignment N = 55 Location: United States	Ad libitum food intake (Time frame: within 15 min of completion of the ad libitum period), hedonic ratings (Time frame: within 5 min prior to ad libitum period), change in blood oxygen (Time frame: 2 weeks) [186]
Bromocriptine (1.25 mg/day during the first week and 2.5 mg/day during the second week)	Obesity	Interventional, randomized, single group assignment N = 8 Location: Netherlands	Difference in 18F-fluorodeoxyglucose uptake, in energy expenditure, in core body temperature and in in insulin sensitivity before and after using bromocriptine (Time frame: 17 months) [187]
Cabergoline (0.5 mg twice weekly)	Body weight Glucose tolerance	Interventional, randomized, parallel assignment N = 40 Location: United States	Body weight and glucose (Time frame: 16 weeks) [188]
Fibromyalgia			
Bromocriptine (single dose of bromocriptine 1.25 mg)	Fibromyalgia	Interventional, randomized, crossover assignment N = 100 Location: Switzerland	Brain metabolites (Time frame: Only in sub study 1: 12 to 30 min) Blood oxygen level dependent (BOLD) responses (Time frame: 12 to 45 min) Sensory and emotional pain responses (Time frame: 12 to 20 min) [189]
Rotigotine (DA agonist; 4 mg/24 h)	Fibromyalgia	Interventional, randomized, parallel assignment N = 230 Location: United States	Change from baseline in average daily pain score to the last 2 weeks of the 12-week treatment phase change from baseline in average daily pain score to the last 2 weeks of the 12-week treatment phase (Based on the Per Protocol Set) (Time frame: baseline, last 2 weeks of the 12-week treatment phase) [190]
Ropinirole	Fibromyalgia	Interventional, randomized, parallel assignment N = 164 Locations: Belgium, Denmark, Finland, France, Germany, Italy, Netherlands, Sweden, United Kingdom	Change in pain intensity score from baseline to last week of treatment (week 12). Pain severity and impact on physical function, sleep quality, tender point pressure threshold [191]
Pramipexole (0.75 mg to 4.5 mg tablets once daily in the evening)	Fibromyalgia	Interventional, randomized, parallel assignment N = 61 Location: United States	Change in the weekly mean of the 24 h average pain score from a daily diary as measured by the 11-point Likert pain scale (Time frame: week 29) [192]
Blood pressure			
Fenoldopam (0.05 μg/kg/min for 3 h)	Hypertension	Interventional, randomized, crossover assignment N = 44 Location: United States	Urine sodium excretion (Time frame: 7 days) [193]
Fenoldopam (0.5 μg/kg/min for 3 h)	Salt-sensitive hypertension	Interventional, randomized, crossover assignment N = 45 Location: United States	Urinary sodium excretion (Time frame: 3 h) [194]
DA (beginning at 5 μg/kg/min and titrated by 5 μg/kg/min to effect up to maximum of 20 μg/kg/min)	Hypotension	Interventional, randomized, parallel assignment N = 70 Location: United States	Number of subjects in each group who have achieved an optimal mean blood pressure value at 24 h of life (Time frame: 24 h) [195]
Cancer			
Cabergoline (total week dose of 3.5 mg, starting 6 months after of ranssphenoidal surgical)	Pituitary adenoma	Interventional, randomized, single group assignment N = 140 Location: Brazil	Tumor shrinkage (time frame: 24 months) [196]
Ropirinole (0.25 mg/day–6.0 mg/day oral)	Prolactinoma	Interventional, single group assignment N = 16 Location: United States	Percentage of subjects that achieved stable prolactine normalization (Time frame: 6–12 months) [197]
Cabergoline (twice weekly for weeks 1 to 4. Courses repeat every 4 weeks in the absence of disease progression or unacceptable toxicity)	Breast cancer	Interventional, single group assignment N = 20 Location: United States	Overall Response Rate at 2 Months [198]
Others			
Cabergoline (1.0, 2.0 and 3.5 mg/week)	Acromegalia	Observational, case only, prospective N = 19 Location: Brazil	IGF-I, GH and prolactin levels (Time frame: 6 months) [199]
Cabergoline	Adverse reaction to other drugs and medicines	Interventional, randomized, single group assignment N = 48 Location: Turkey	Effect of prolactin vascular flow and resistance (Time frame: the effect of prolactin in vascular resistance at 2 weeks after treatment) [200]

## Data Availability

Not applicable.

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
