# Peer review of "Therapeutic Potential of Dopamine and Related Drugs as Anti-Inflammatories and Antioxidants in Neuronal and Non-Neuronal Pathologies"

_pharmaceutics, 2023, doi:10.3390/pharmaceutics15020693_

Round 1

Reviewer 1 Report

Following the analysis of the manuscript titled "Therapeutic potential of dopamine and related drugs as anti-inflammatory, antioxidant, in non-neuronal pathologies", I appreciate the article's topic is interesting and the presentation of the information is clear and properly structured. I recommend that it should be revised taking into account the following observations:

-          Abstract: please define the type of manuscript and insert Results data.

-          Introduction: Provide several recent references for "DA and dopaminergic drugs bromocriptine, cabergoline, domperidone, fenoldopam, pramipexole, ropinirole, quinagolide show anti-inflammatory and antioxidant functions."

-          Before Conclusions, clarify the strengths and the limitations of this narrative review, if any.

-          Carefully check throughout the manuscript for missing words (e.g. abstract - Dopamine and dopaminergic drugs can a viable option for additional treatments to sepsis...)  or the ones that should be deleted (e.g. page 10 - SUKANOVA DAJAS GASSEN HARA).

-          Please provide an abbreviations list at the end of the manuscript.

Author Response

Comments and Suggestions for Authors

Following the analysis of the manuscript titled "Therapeutic potential of dopamine and related drugs as anti-inflammatory, antioxidant, in non-neuronal pathologies", I appreciate the article's topic is interesting and the presentation of the information is clear and properly structured. I recommend that it should be revised taking into account the following observations:

Dear reviewer, thank you for your valuable comments. You could find the performed changes in the new manuscript.

  1. Abstract: please define the type of manuscript and insert Results data.

R= Following your suggestions, we added the type of manuscript and we inserted results data.

  1. Introduction: Provide several recent references for "DA and dopaminergic drugs bromocriptine, cabergoline, domperidone, fenoldopam, pramipexole, ropinirole, quinagolide show anti-inflammatory and antioxidant functions."

R= We added recent references for DA and dopaminergic drugs as you suggested.

  1. Before Conclusions, clarify the strengths and the limitations of this narrative review, if any.

R=We added the strengths and the limitations of this review as you required.

  1. Carefully check throughout the manuscript for missing words (e.g. abstract - Dopamine and dopaminergic drugs can a viable option for additional treatments to sepsis...) or the ones that should be deleted (e.g. page 10 - SUKANOVA DAJAS GASSEN HARA).

R=We checked and corrected the manuscript as you recommended.

  1. Please provide an abbreviations list at the end of the manuscript.

R=We added the abbreviations list at the end of the manuscript as your required.

Reviewer 2 Report

Dear Editor; The at tached article was checked. The manuscript contains interesting information about the Therapeutic potential of dopamine and related drugs as anti-inflammatory, antioxidant, in non-neuronal pathologies

 I think that this article is well suits to your journal. This article would draw many citations.

It is generally good to work. The scientific and presentation level of the manuscript is high. 

 -In the Introduction section, the authors should state the objectives of the work, define the scope of your paper, summarize relevant work to the study being reported and provide an adequate background. The introduction must be designed to inform the reader of the rationale and significance of the study." The aim of the study and relevant work should be stated. This section is very insufficient.

 -before dopamine…Herbs are used for antioxidant purposes. That herbs are used for antioxidant purposes may be added in the introduction part or in the discussion part of the article.

 Uba, A.I.; Zengin, G. Montesano, D., Cakilcioglu, U., Selvi, S., Ulusan, M.D., Caprioli, G., Sagratini, G., Angeloni, S., Jugreet, S.  Hasan, M.M., Mahoomodally, M.F., 2022. Antioxidant and enzyme ınhibitory properties, and HPLC–MS/MS profiles of different extracts of Arabis carduchorum Boiss.: An endemic plant to Turkey. Applied Sciences, 12, 6561 doi.org/10.3390/app12136561

Khatun, S., Chatterjee, N.C., Cakilcioglu, U., (2011). Antioxidant activity of the medicinal plant Coleus forskohlii Briq. African Journal of Biotechnology, 10(13), 2530-2535. Doi: 10.5897/AJB10.2526

-Figures are very well prepared, please cite if taken from another source.

-Methodology is intelligible

-In method: (…..as well as internationally recognized databasesGoogle Scholar, Web of Science, SciFinder…. Have you searched?

-References were cross-checked.

-In referencer 168-198; Do not leave spaces between words

-The paper should be edited according to the writing rules of the journal

Author Response

Comments and Suggestions for Authors

Dear Editor; The attached article was checked. The manuscript contains interesting information about the Therapeutic potential of dopamine and related drugs as anti-inflammatory, antioxidant, in non-neuronal pathologies.  I think that this article is well suits to your journal. This article would draw many citations. It is generally good to work. The scientific and presentation level of the manuscript is high. 

Dear reviewer.  We are grateful for your valuable comments. You could fine the performed changes in the new manuscript.

  1. In the Introduction section, the authors should state the objectives of the work, define the scope of your paper, summarize relevant work to the study being reported and provide an adequate background. The introduction must be designed to inform the reader of the rationale and significance of the study." The aim of the study and relevant work should be stated. This section is very insufficient.

R= We improve the introduction writing as you recommended, we agree with you, and we consider all the points that your mention.

  1. Before dopamine…Herbs are used for antioxidant purposes. That herbs are used for antioxidant purposes may be added in the introduction part or in the discussion part of the article.

 Uba, A.I.; Zengin, G. Montesano, D., Cakilcioglu, U., Selvi, S., Ulusan, M.D., Caprioli, G., Sagratini, G., Angeloni, S., Jugreet, S.  Hasan, M.M., Mahoomodally, M.F., 2022. Antioxidant and enzyme ınhibitory properties, and HPLC–MS/MS profiles of different extracts of Arabis carduchorum Boiss.: An endemic plant to Turkey. Applied Sciences, 12, 6561 doi.org/10.3390/app12136561

Khatun, S., Chatterjee, N.C., Cakilcioglu, U., (2011). Antioxidant activity of the medicinal plant Coleus forskohlii Briq. African Journal of Biotechnology, 10(13), 2530-2535. Doi: 10.5897/AJB10.2526

R=We added in the introduction the information about related herbs as you suggested. We totally agree with you, is important to highlight the antioxidant properties of herbs in our review.

  1. Figures are very well prepared, please cite if taken from another source.

R= We really appreciated your comment, the figures are original and they were elaborated by our team.

  1. Methodology is intelligible.

R= We appreciated your comment.

  1. In method: (…..as well as internationally recognized databases Google Scholar, Web of Science, SciFinder…. Have you searched?

R=Yes, we have searched in all databases that we considered in the methodology part, including Google Scholar and Web of Science (we added Web of Science because we used this tool). However, we did not search in SciFinder, we did not know this database, thanks for your recommendation.

  1. References were cross-checked. In referencer 168-198; Do not leave spaces between words

R= We change these references according to your recommendation.

  1. The paper should be edited according to the writing rules of the journal

R= We performed this recommendation very carefully.

Reviewer 3 Report

The authors did a review on "Therapeutic potential of dopamine and related drugs as anti-inflammatory, antioxidant, in non-neuronal pathologies". Some of the minor comments for improvement are

1. Introduction is very short, add some more information about applications.

2. Follow the same formatting for table 1 and 2

3. What is the novelty of this review paper.

Author Response

Comments and Suggestions for Authors

The authors did a review on "Therapeutic potential of dopamine and related drugs as anti-inflammatory, antioxidant, in non-neuronal pathologies". Some of the minor comments for improvement are:

  1. Introduction is very short, add some more information about applications.

R= Dear reviewer, we appreciate your important comments, we agree with you about this recommendation, we improved the introduction content.

  1. Follow the same formatting for table 1 and 2

R= We homogenize the format of the two tables as you suggested.

  1. What is the novelty of this review paper?

R= We added the novelty of this review in the conclusion. We believe that the repositioning of DA and dopaminergic drugs could impact in new neuronal and non-neuronal pathologies because DA has anti-inflammatory and antioxidant properties. We described the main biochemical mechanisms that are related to these benefits with the aim to propose them in pathologies in which patients require a reduction of inflammatory and oxidative response. We expected that with this review the interest and the impulse of further clinical trials will increase.

Reviewer 4 Report

The introduction requires a deep explanation of the objective and related studies

The methodology must be followed index books

add some natural drugs (herbal bioactive compounds) potential of dopamine

Author Response

Comments and Suggestions for Authors

  1. The introduction requires a deep explanation of the objective and related studies

R=Dear reviewer, we are grateful for your important comments. We improved the introduction as you suggested.

  1. The methodology must be followed index books

R= We consider your suggestion.

  1. Add some natural drugs (herbal bioactive compounds) potential of dopamine.

R= We totally agree with you, we added information related to your important comment in the introduction, point 5 and discussion and conclusions sections.

Reviewer 5 Report

This study represents a review article related  to  the anti-inflammatory, antioxidant potential of dopamine and derivatives drugs  in non-neuronal pathologies.

This research is important and can bring valuable information with practical application, and that could be important for the future research. The presented research is well-planned, and the manuscript is well organized.

Therefore, the work could be of interest, but some points must be reconsidered prior acceptance:

The title draws our attention especially to the antioxidant and anti-inflammatory potential, and the authors pay more attention to actions related to neuronal pathologies. The title should be changed to reflect this aspect. These actions can influence and explain the therapeutic response in neuronal pathology....?

More information should be included about the antioxidant and anti-inflammatory potential of the DA natural derivatives and of the natural vegetable sources of these compounds.

Abbreviations should be corrected - numerical indices with subscripts.

Starting from specifying the deficient areas of this approach, the authors should mention the research directions that could be approached in perspective to complete the information in this field. These could be specified in the conclusions.

Author Response

Comments and Suggestions for Authors

This study represents a review article related to the anti-inflammatory, antioxidant potential of dopamine and derivatives drugs in non-neuronal pathologies. This research is important and can bring valuable information with practical application, and that could be important for the future research. The presented research is well-planned, and the manuscript is well organized.

Therefore, the work could be of interest, but some points must be reconsidered prior acceptance:

Dear reviewer, we really appreciated your important comments, you could find the changes in the new manuscript

The title draws our attention especially to the antioxidant and anti-inflammatory potential, and the authors pay more attention to actions related to neuronal pathologies. The title should be changed to reflect this aspect.

R= We modified the title as you suggested.

These actions can influence and explain the therapeutic response in neuronal pathology....?

R= The anti-inflammatory and antioxidant properties, among other effects, of DA and dopaminergic drugs also influence neuronal diseases, however, the need to increase dopamine levels stands out in pathologies where its deficiency is the main mechanism of damage.

More information should be included about the antioxidant and anti-inflammatory potential of the DA natural derivatives and of the natural vegetable sources of these compounds.

R=We agree with you, we included information in the introduction, in point 5 and in the conclusion sections.

Abbreviations should be corrected - numerical indices with subscripts.

R= We performed these changes.

Starting from specifying the deficient areas of this approach, the authors should mention the research directions that could be approached in perspective to complete the information in this field. These could be specified in the conclusions.

R= We improved the conclusions including your requirements.

Reviewer 6 Report

I have gone through the manuscript. Topic appears to be interesting but it needs conditioning. Authors have highlighted the role of dopamine in prevention and amelioration of different diseases. However, authors did not comprehensively describe a detailed analysis of cell signaling pathways. Similarly, dopamine signaling mediated downstream effectors are not explained contextually in different diseases. 

Dopamine mediated cancer chemopreventive role also needs to be addressed in this issue. How dopamine played role in the regulation of carcinogenesis and metastasis. 

Author Response

Comments and Suggestions for Authors

 I have gone through the manuscript. Topic appears to be interesting, but it needs conditioning

  1. Authors have highlighted the role of dopamine in prevention and amelioration of different diseases. However, authors did not comprehensively describe a detailed analysis of cell signaling pathways. Similarly, dopamine signaling mediated downstream effectors are not explained contextually in different diseases. 

R= Dear reviewer, we really appreciate your comments. The changes in the manuscript are in color green.  We added information related to signaling pathways as you recommend, and we describe with more details the biochemical mechanism in the figure 3 legend.

  1. Dopamine mediated cancer chemopreventive role also needs to be addressed in this issue. How dopamine played role in the regulation of carcinogenesis and metastasis.

R=We added the information as you suggested (Table 2 and section 5). We totally agree with you, DA and dopaminergic drugs has potential in cancer, but we only describe the anti-inflammatory and antioxidant effect of this compounds because, in oncological diseases,  DA through D2 receptor could be an important antiangiogenic drug in many types of cancer, but we are writing another article taking account this important mechanism.

Round 2

Reviewer 2 Report

The attached articled was checked. The author made necessary corrections. The manuscript can be published in this form.

Regards

Reviewer 4 Report

Accept

Reviewer 6 Report

It looks in GOOD form now.